# Critical Factors Affecting Fire Safety in High-Rise Buildings in the Emirate of Sharjah, UAE

**Musab Omar \*** **, Abdelgadir Mahmoud and Sa'ardin Bin Abdul Aziz**

Razak Faculty of Technology and Informatics, Universiti Teknologi Malaysia, Kuala Lumpur 54100, Malaysia
\* Correspondence: alkhaldy70@hotmail.com; Tel.: +97-1527771351

**Abstract:** The purpose of this paper is to identify the critical factors affecting fire accidents in high-rise residential buildings in the Emirate of Sharjah in order to find solutions that contribute to reducing injuries and deaths from fire accidents. A large urban expansion of the Emirate of Sharjah has taken place in the form of high-rise buildings, and the Emirate is now third in the UAE in terms of the number of high-rise buildings and is home to 19% of the population. As a consequence, an increase in the rate of fire accidents has also been observed. As such, there is a need to conduct research on enhancing fire safety in high-rise buildings by conducting a literature review, in which nineteen factors affecting fire globally were identified. Because the fire characteristic is unique in every country, to identify the nature of fire in the Emirate of Sharjah, we consulted sixteen subject matter experts in the field of fire in the Emirate of Sharjah to identify the factors applicable to the Emirate. We used the failure mode, effect, and criticality analysis methodology to accomplish this goal. The outcome of the consultations resulted in the three main factor categories, which are management factors, human factors, and technical factors, and the critical factors affecting the high-rise buildings in the Emirate of Sharjah were identified, which are: fire regulations, fire enforcement regulations, accident investigation, rescue speed, human behavior, lack of proper maintenance, fire training, building design, fire knowledge, combustible materials, fire culture of society, and urbanization. Using the Analytical Hierarchy Process (AHP) tools implemented to measure the effect level of the sub-critical fire factors, 45 effects were identified, and the most common effects were: the building is fully covered by cladding, the effect of stopping activities in HRBs that are non-compliant with fire regulations, the residents practice activities related to fire knowledge, fire regulations efficiency, the training of new employees by their employers, and the residents have fire-related knowledge.

**Keywords:** fire factor; residential buildings; fire accidents



## 1. Introduction

With the development of urbanization and the growth of complex industries in the Emirate of Sharjah, the fire accident rate increased in the Emirate of Sharjah compared to those in other Emirates inside the UAE for the period from 2013 to 2018, according to data published in the UAE Ministry of Interior report of 2019. Fire accidents are a real problem that should be addressed to avoid them affecting society in the Emirate of Sharjah. Fire accidents will affect the economy of the Emirate of Sharjah because they have a direct effect on the real estate market, industrial activities, and business reputation, eventually leading to a decrease in the competitiveness of the Emirate of Sharjah in the region.

High-rise buildings and very tall buildings have dramatically increased in number, and consequently, the number of factors that affect the cause and/or development of fire has also increased. It is difficult to quantify the factors, and they are not independent from each other. A degree of ambiguity exists, so fire-related problems in high-rise building have become a worldwide concern [1]. Fire safety studies are of great significance in improving our understanding of the nature of fire phenomena and how fires develop. As such, it is necessary that we carry out fire prevention and control measures [2]. Firefighters can

accurately predict the places, types, and regional distribution of potential fire hazards, and they can focus on the seasons and populations prone to fire disasters [3]. The definition of a fire disaster here is a fire that is burning out of control in a space over time. Fire disasters have become one of the most destructive disasters in modern society due to their high frequency and serious destructiveness [4]. Buildings are major sources of urban fires; thus, fire prevention training programs should be provided, particularly to those in densely populated urban areas [5]. The development of fire safety strategies should be a continuous process such that fire safety systems are regularly reviewed and maintained [6]. The fire safety framework involves the enhancement of fire safety in four key areas: fire protection features in buildings; regulation and enforcement; consumer awareness; technology and resource advancements [7]. Effective fire safety management is a critical task in the planning, design, and operation of a building; furthermore, the occupants/users of a building should be familiar with the escape routes in case of a fire, and maintenance staff must be provided with the relevant information about responsible staff, key locations, and fire safety equipment to ensure that the equipment is in good working order [8]. The failure of all of the alarm and extinguishing systems can accelerate the progress of the fire and hinders people's awareness of the accident and their timely response. Therefore, it is necessary to develop a safe environment that allows more time for people to leave a risky place [9]; the influencing factors on the high-rise building fires are related to people, objects, environment, technology, and management [10]. A study in Nigeria indicated that the most common causes of fire incidents in high-rise buildings are electrical faults [11]. Crowd evacuation in high-rise buildings in case of a fire becomes a major safety issue. In a fire environment, personnel evacuation behavior in high-rise buildings shows complex multi-directional characteristics [12]. Fire load and heat release rate are important considerations during a fire. In order to assess the fire risk of high-rise residential buildings, possible fire scenarios should be identified. There is an urgent need to collect data on the fire load and identify the heat release rate for this type of building [13]. High-rise building fires have many characteristics, such as the diversity of the blazes, factors affecting them, various ways of the fires spreading, and the difficulty of evacuation [14].

The meaning of high-rise buildings (HRBs) varies from one country to another. According to the NFPA, a high-rise building is "A building where the floor of an occupiable story is greater than 75 ft (23 m) above the lowest level of fire department vehicle access". In China, residential buildings with seven stories or more are defined as high-rise buildings. They could be further subdivided into middle-high-rise, high-rise, and super high-rise buildings according to their number of stories and height. The term "high-rise building" in Korea is defined as a reinforced concrete structure with 30 stories or more [15].

In the Emirate of Sharjah, UAE, the definition of a high-rise building is "The occupancies or Multiple and Mixed occupancies, facilities, buildings and structures having total height of the building (excluding roof parapets) is between 23 m to 90 m from the lowest grade or lowest level of Fire Service access into that occupancy", and the definition of a super high-rise building is "The occupancies or Multiple and Mixed occupancies, facilities, buildings and structures having total height of the building (excluding roof parapets) is more than 90 m from the lowest grade or lowest level of Fire Service access into that occupancy" [16].

The Emirate of Sharjah is third in the UAE in terms of the number of high-rise buildings it has, and it is classified as one of the Emirates with rapid urban growth. The UAE ranks in third place in the world in terms of the number of skyscrapers it has, and it is home to 251 buildings that are taller than 150 m. As the safety of high-rise buildings is a global concern, in this study, we review the factors that affect the fire prevention systems of HRBs and super high-rise buildings in the Emirate of Sharjah.

The region of UAE has dry, subtropical weather with year-round sunny days and rare, shallow rainfall. The weather is extremely hot and humid along the shoreline. The summers from the months of June to September are extremely hot and humid, with temperatures reaching 48 °C (118 °F) and the humidity being as high as 80–90% [17]. The Emirate of

Sharjah is considered to be the third largest Emirate in the UAE in terms of area, which covers 2600 km$^2$; 19% of the UAE population live in Sharjah, and the Emirate is home to people of 200 different nationalities. Moreover, 1.5 million tourists visit the Emirate of Sharjah annually.

The contribution of this paper is to identify the critical factors that affect the current fire management system implemented in high-rise buildings in the Emirate of Sharjah, the first step in correction, and the diagnostic procedure and to determine the area of failure, which are provided in detail in this paper through the fire effect weight listed in the sub-critical factors effecting the fire management in HRBs in the Emirate of Sharjah.

## 2. Methods

By reviewing the literature related to fire in HRBs, 15 research papers were reviewed and the factors affecting the safety of HRBs from fire risks were monitored, and based on the frequency of factors in the reviewed papers, 20 factors affecting the fire system in HRBs were identified. These factors must have been applicable to the Emirate of Sharjah, and therefore, 16 experts in the field of fire in the Emirate of Sharjah were consulted; their experience spans more than 10 years, and they were identified as stakeholders in Civil Defense, the Prevention and Safety Authority, maintenance and installation companies, and distribution companies. The Failure Mode Effect and Criticality Analysis (FMECA) tool was used to analyze the pattern affecting the identified factors directly related to the Emirate of Sharjah, including fire equipment, fire equipment factories, and fire systems designed for offices. After determining the factors, an analysis was performed for each factor separately to determine the possible failure patterns by calculating their severity, occurrence, and means of detection, and the factors that were evaluated as being very important were considered as critical factors that affect HRBs in the Emirate of Sharjah. The data of critical factors were used to develop a fire factor effect index for high-rise buildings based on the subject matter experts through the application of the Analytical Hierarchy Process (AHP) tools to measure the level of fire sub factor effect on the high-rise buildings, and the method was implemented as shown in Figure 1.

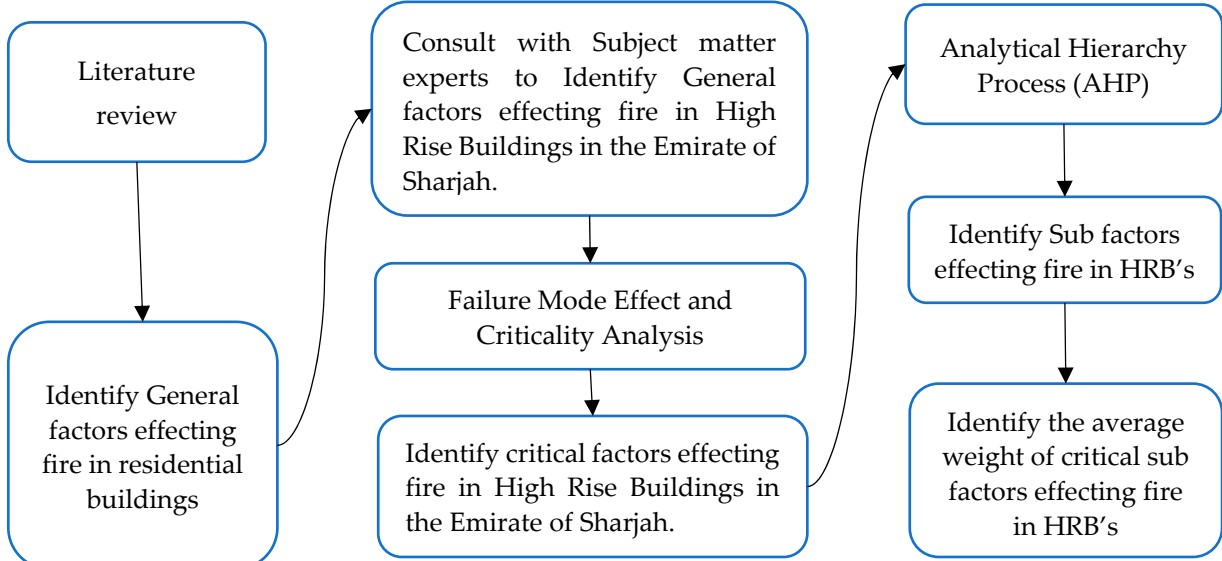

**Figure 1.** Method of Research.

## 3. Results

### 3.1. Fire Factors Effecting Fire Management System in the Residential Buildings

After reviewing research and studies that dealt with the fire factors in residential buildings, and after reviewing 15 journals related to fire as shown in Table 1, 19 factors affecting fire in residential buildings were identified, as shown in Table 2. It is not necessarily the

case that all of these factors affect the Emirate of Sharjah. Each region has a different kind of fire, which depends on the weather, laws, materials used in construction, fire culture, and the compliance of residential buildings with fire regulations, so we deemed it necessary to present the factors identified from published research to experts in the Emirate of Sharjah in order to determine the general factors affecting high-rise residential buildings in the Emirate. Specifically, and based on the review results, four main categories of factors were identified: administrative factors, technical factors, human factors, and other factors, as specified in Table 3.

**Table 1.** Reviewed papers.

| [18] | [19] | [20] | [4] | [21] |
|------|------|------|-----|------|
| [7]  | [22] | [23] | [24] | [8] |
| [5]  | [25] | [26] | [6] | [27] |

**Table 2.** Factors effecting fire management system in the residential buildings.

| No. | Factors | Weight |
|-----|---------|--------|
| 1.  | Building design | 5% |
| 2.  | Fire regulations | 6% |
| 3.  | Facilities management and policies | 5% |
| 4.  | Rescue speed | 6% |
| 5.  | Fire knowledge | 6% |
| 6.  | Fire equipment | 6% |
| 7.  | Human behavior | 5% |
| 8.  | Firefighting maintenance | 5% |
| 9.  | Fire culture of society | 5% |
| 10. | Fire training | 6% |
| 11. | Combustible materials | 6% |
| 12. | Fire enforcement regulations | 6% |
| 13. | Fire data analysis/availability | 4% |
| 14. | Accident investigation | 6% |
| 15. | Fire R&D | 4% |
| 16. | Fire technology | 5% |
| 17. | Public/contractor attitude | 5% |
| 18. | Staff assignment | 5% |
| 19. | Climate change | 4% |

Administration factors such as fire regulation, rescue speed, fire regulation enforcement, and accident investigation management have the highest frequency in the literature review, while human factors such as fire training and fire knowledge have the highest frequency, and technical factors such as fire equipment and combustible material counted in are at the top of the list.

The general factors affecting HRB fire management systems that were identified from the literature review were sent to experts in the field of fire protection in the Emirate of Sharjah who are stakeholders in the fire management system such as: government authorities, firefighting installation and maintenance contractors, fire consultant offices, or fire agents and distributers, and the results are shown below. The three factors identified by the subject matter experts are the government structure factors, as well as the urban

planning and urbanization factors affecting the efficiency of fire prevention management systems in the Emirate of Sharjah. As shown in Table 3, based on the Delphi technique method, the third round of factors achieving 75% of the subject matter expert consensus are identified as factors affecting the Emirate of Sharjah.

**Table 3.** General factors effecting the fire management system in the Emirate of Sharjah.

| No. | Factors | Weight |
|---|---|---|
| 1. | Building design | 6% |
| 2. | Fire regulations | 7% |
| 3. | Facilities management and policies | 2% |
| 4. | Rescue speed | 7% |
| 5. | Fire knowledge | 9% |
| 6. | Fire equipment | 2% |
| 7. | Human behavior | 5% |
| 8. | Firefighting maintenance | 5% |
| 9. | Fire culture of society | 8% |
| 10. | Fire training | 5% |
| 11. | Combustible materials | 9% |
| 12. | Fire enforcement regulations | 9% |
| 13. | Accident investigation | 9% |
| 14. | Fire technology | 2% |
| 15. | Public/contractor attitude | 8% |
| 16. | Urbanization | 4% |
| 17. | Government structure | 3% |
| 18. | Urban planning | 1% |
| 19. | Resource allocation | 6% |

The top factors ranking ones are: fire knowledge, fire enforcement regulations, combustible materials, accident investigation, public/contractor attitude, fire culture of society, fire regulations, and rescue speed, which reflect the general fire factors affecting the high-rise buildings in the Emirate of Sharjah, and the general factors need to be evaluated to identify the critical factors affecting the HRBs fire safety by using the failure mode, effect, and criticality analysis (FMECA).

*3.2. Failure Mode, Effect, and Criticality Analysis*

To determine the critical factors affecting fires in high-rise residential buildings in the Emirate of Sharjah, FMECA was used. Failure mode, effect, and criticality analysis (FMECA) is one of the most robust and widely implemented engineering risk management tools. To enhance its applicability of addressing the different aspects of engineering problems, FMECA is often integrated with other techniques related to multicriteria decision-making (MCDM) processes [28]. The main factors affecting the HRB fire prevention management systems in the Emirate of Sharjah were examined with the help of subject matter experts, and the possible failures in each factor were identified.

The criteria for dealing with failure are classified in Table 4. FMECA descriptions according to the fault type, the degree, and the number of impacts, and they are assessed in terms of severity, occurrence, and detection. In terms of severity and occurrence, one represents the least impacted one, and ten represents the most impacted one. In terms of detection, one represents a defect being detectable, and ten represents a defect being non-detectable. The criteria were validated by the subject matter experts.

**Table 4.** FMECA descriptions.

| Degree | Number | Severity |
|---|---|---|
| low | 1 | The defect is limited and cannot affect the effectiveness of the fire prevention management system |
|  | 2 |  |
|  | 3 |  |
| Medium | 4 | It can cause controllable failure |
|  | 5 |  |
|  | 6 |  |
| High | 7 | It can weaken the fire protection system |
|  | 8 |  |
|  | 9 |  |
|  | 10 |  |
| Degree | Number | Occurrence |
| low | 1 | The defect applies to only a few parts of the system |
|  | 2 |  |
|  | 3 |  |
| Medium | 4 | The defect applies to 50% or more of the system components |
|  | 5 |  |
|  | 6 |  |
| High | 7 | The defect applies to more than 75% of the components of the fire fighting system |
|  | 8 |  |
|  | 9 |  |
|  | 10 |  |
| Degree | Number | Detectability |
| High | 1 | There is a possibility of identifying the defect |
|  | 2 |  |
|  | 3 |  |
| Medium | 4 | There is a possibility of us not being able to identify the defect |
|  | 5 |  |
|  | 6 |  |
| Low | 7 | There is a high probability of us not being able to identify the defect |
|  | 8 |  |
|  | 9 |  |
|  | 10 |  |

The risks involved in a fire prevention system are completely dependent on the defect severity, but severity is not the only influencing factor that determines the critical situation of a failure. The possibility of a fault occurring is an important factor, but the possibility of detecting the fault is the most important and influential factor, as the possibility of detection determines the possibility of controlling the malfunction. In Table 5. FMECA rules, samples of the basis of the risk assessment are given, and the relevant criteria are specified, which involves a combination of impact severity, the possibility of defect occurrence, and the possibility of defect detection. The faults classified by the experts into the medium- and high-severity groups with the possibility of medium- and high-severity occurrences when the possibility of detection is low are treated as very important. Moreover, if the possibility

of detection is medium, it is addressed on the basis that it is important. The basis for the risk evaluation was validated by the subject matter experts.

**Table 5.** FMECA rules.

| Severity | Occurrence | Detection | Risk |
|---|---|---|---|
| Medium | Medium | Low | Very Important |
| Medium | Medium | Medium | Important |
| High | Medium | Low | Very Important |
| High | Medium | Medium | Important |
| High | High | Low | Very Important |
| High | High | Medium | Important |

*3.3. Critical Success Factors Affecting the Fire Prevention Management Systems in the Emirate of Sharjah*

Based on the assessment of subject matter experts in the field of HRB fire prevention systems in the Emirate of Sharjah, the severity of the impact of a potential defect was evaluated for each of the factors that were previously identified as those affecting HRBs in the Emirate of Sharjah. The probability of occurrence and the possibility of detection were evaluated, and the majority of the experts agreed on the evaluation according to Table 6, in which the types of defects are listed in order of importance, ranging from very important to important to non-important. Accordingly, 91 very important faults, 62 important faults, and 6 non-important faults were identified, as shown in Figure 2.

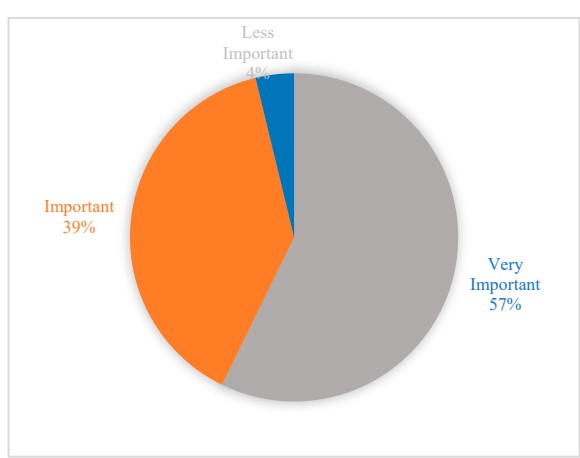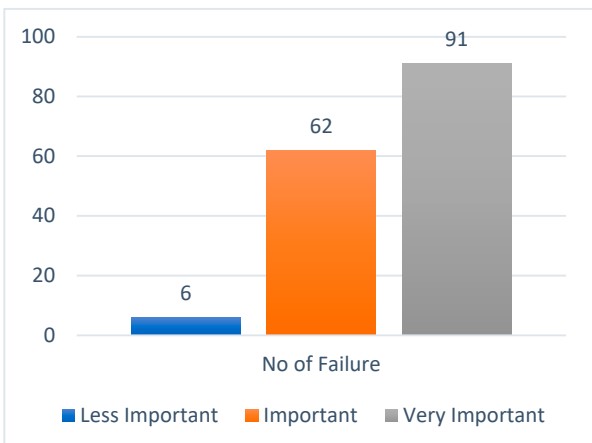

**Figure 2.** FMECA outcome.

The failure in each factor was identified, as shown in Table 6, and after the analysis according to the sample of the basis for risk evaluation, the degree of importance of each failure was determined based on the risks it contains. Some of the factors for which the failures were evaluated were of a high risk, so they were rated as being very important. These factors were classified as critical in the HRB fire prevention systems in the Emirate of Sharjah.

Based on the number of failures rated as very important in each factor, 12 factors were observed to have a high frequency of very important ratings, as shown in Table 7. Critical factors effecting the fire management in the HRBs in the Emirate of Sharjah. According to the very important failure mode frequency, the critical factors affecting HRB fire management systems have been identified.

**Table 6.** Failure mode effect criticality analysis.

| Factor: Fire Regulations | S | O | D | Risk |
|---|---|---|---|---|
| It does not include all types of residential buildings. | M | M | L | V.Imp |
| Do not support continuous improvement in the fire prevention management system. | M | M | M | Important |
| Do not comply with the applicable procedures. | M | M | H | Not Imp |
| Does not support research and development. | H | M | L | V.Imp |
| The investigation procedures in fire accidents are not required or specified. | H | H | L | V.Imp |
| It did not specify the mechanisms and procedures for response and rescue. | H | M | L | V.Imp |
| It does not support optimum utilization of resources. | M | H | L | V.Imp |
| The requirements of the administrative structure did not specify the fire protection system at the level of the authority and stakeholders. | H | M | L | V.Imp |
| It did not specify the procedures required to educate people about the fire. | H | L | L | Not Imp |
| Do not request reporting of near misses or fire incidents that did not require the intervention of firefighters. | H | H | L | V.Imp |
| There are no procedures in the legislation regulating the investigation of fire accidents. | H | H | L | V.Imp |
| **Factor: fire enforcement regulations** | **S** | **O** | **D** | **Risk** |
| The process of inspection of residential building is not carried out in a regular periodic manner. | H | M | L | V.Imp |
| Audit for licensed entities not carried out periodically. | H | M | L | V.Imp |
| The process of issuing certificates of compliance is not based on precise, specific, and strict criteria. | H | M | L | V.Imp |
| The process of issuing certificates of completion is not based on precise, specific, and strict criteria. | H | M | L | V.Imp |
| Licensing processes for companies are not based on precise, specific, and strict criteria. | H | H | L | V.Imp |
| Failure to take the necessary measures against the procedures related to residential establishments that violate. | H | M | L | V.Imp |
| Not to take punitive measures for licensed companies that violate legislation. | H | H | L | V.Imp |
| Failure to check the qualification requirements of employees of licensed companies. | H | H | L | V.Imp |
| Lack of focus on obligating high-rise buildings. | H | M | L | V.Imp |
| **Facilities Management and policies** | **S** | **O** | **D** | **Risk** |
| Lack of safety management policies or procedures. | H | H | M | Important |

| Fire Training | S | O | D | Risk |
|---|---|---|---|---|
| Workers not receiving fire fighting training. | H | M | L | V.Imp |
| Ineffective training. | H | M | L | V.Imp |
| Training of employees is not mandatory. | H | M | L | V.Imp |
| There are no mechanisms to achieve mandatory training | H | M | L | V.Imp |
| Resetting the system and stopping the alarm by the building guard without verifying the fire. | H | H | M | Important |
| Shutdown the system completely by disconnecting the electrical power source in the event of a frequent bell. | H | M | M | Important |
| Fire pumps are in the off position. | H | M | L | V.Imp |
| Fire pumps are isolated from power. | H | M | M | Important |
| Diesel pump without fuel. | H | M | L | V.Imp |
| The starting batteries in the diesel pump are disconnected or not working. | H | M | M | Important |
| The water tank is not full according to the design capacity of the fire extinguishing system. | H | M | L | V.Imp |
| Use the pump room as a material store room. | M | M | M | Important |
| Lack of knowledge of dealing with fire alarm panel and other extinguishing equipment. | H | M | L | V.Imp |
| The use of fire hoses to wash the corridors. | H | M | M | Important |
| **Fire Knowledge** | **S** | **O** | **D** | **Risk** |
| Residents lack knowledge of fire hazards. | H | H | L | V.Imp |
| Residents lack knowledge of fire behavior and its causes. | H | H | L | V.Imp |
| Residents lack knowledge of the procedures required in the event of a fire. | H | H | L | V.Imp |
| Residents lack knowledge of when to use fire equipment. | H | H | L | V.Imp |
| **Fire Society culture** | **S** | **O** | **D** | **Risk** |
| The community's religious culture of predestination does not support the possibility of avoiding fire accidents. | M | M | L | V.Imp |
| The culture of the community about the causes of fire varies among the residents of the same residential establishment. | M | M | L | V.Imp |
| The culture of the community does not support taking preventive measures against fire hazards. | M | M | L | V.Imp |

**Table 6.** *Cont.*

| Factor: Fire Regulations | S | O | D | Risk | Fire Training | S | O | D | Risk |
|---|---|---|---|---|---|---|---|---|---|
| No fire risk assessment is done. | H | H | L | V.Imp | The nature of the community in residential establishments Awareness of the consequences that can occur if a fire occurs in a residential building with large groups of residents. | M | M | M | Important |
| There is no record of fire hazards. | H | H | L | V.Imp | Negative culture about fires, which results from religious or ideological beliefs or racial behaviors. | H | M | L | V.Imp |
| There is no internal inspection or audit. | H | H | M | Important | The culture of the population when hearing the sound of the fire alarm, as it is dealt with on the basis that it is a recurring technical failure, and the response is not performed. | H | H | M | Important |
| There is no emergency management plan for high-risk building. | H | H | M | Important | **Fire Technology** | **S** | **O** | **D** | **Risk** |
| There is no qualified employee who supervises the management of the fire system and risk management. | H | H | M | Important | It is not suitable for the nature of residential establishments in the Emirate of Sharjah. | H | M | L | V.Imp |
| The absence of mandatory requirements for the sustain serviceability of fire protection systems in residential building. | M | M | M | Important | Inefficent. | H | M | L | V.Imp |
| The owners of residential building consider that the resources provided for the management of the fire protection system are a waste of money and time. | H | M | M | Important | They are not certified based on reliable reliability procedures. | M | M | M | Important |
| Real estate companies that manage building do not put fire protection systems among their priorities. | H | H | M | Important | Not related to the latest technologies in the field of fire fighting. | M | M | M | Important |
| HVAC not included in the required preventive maintenance. | H | H | M | Important | Early detection of fires works poorly. | H | M | M | Important |
| Electrical connections are not included in the required preventive maintenance. | H | H | M | Important | The competent authority does not rely on early fire detection data. | H | H | M | Important |
| The elevator system is not connected to the alarm and fire fighting system. | M | M | M | Important | The early fire detection system is not approved by the Federal Fire Authority. | M | H | M | Important |
| Owners of residential building do not care about preventive maintenance. | H | H | L | V.Imp | Early detection of fires does not comply with the requirements and specifications of the Federal Authority. | H | M | M | Important |
| Real estate companies do not care about preventive maintenance. | H | M | M | Important | The technology used in fire detection is unsuitable and has frequent breakdowns. | H | M | M | Important |
| **Accident investigation** | **S** | **O** | **D** | **Risk** | The technology used is not compared to other similar areas that apply good practices. | M | M | M | Important |
| Accidents are not investigated by the relevant fire authority. | H | H | L | V.Imp | The technology used has not been evaluated and tested. | H | M | M | Important |
| The investigation of fire accidents is carried out by the Public Prosecution Office for the purposes of providing evidence to the court, for the purpose of compensation procedures related to insurance, or for lawsuits only. The data of the investigation are confidential. | H | M | L | V.Imp | It does not support Fourth Industrial Revolution technologies such as artificial intelligence, big data, robotics, and the Internet of Things. | M | M | M | Important |
| There is no specialized department in the structure of the competent authority to investigate fire accidents. | H | H | L | V.Imp | **Lack/improper Maintenance** | **S** | **O** | **D** | **Risk** |
| There are no qualified personnel to investigate fire accidents at the competent authority. | H | H | L | V.Imp | Inefficiency of installation and maintenance contractors. | H | H | M | Important |
| Fire accident investigation data are not seen as important data that need to be obtained. | H | H | L | V.Imp | Lack of clarity about the mechanism for reporting alarm and fire fighting system malfunctions to the responsible installation and maintenance contractors. | H | H | M | Important |

**Table 6.** *Cont.*

| Factor: Fire Regulations | S | O | D | Risk |
|---|---|---|---|---|
| Some fire accidents are repeated periodically because fire accidents are not investigated. | H | H | L | V.Imp |
| The root causes of fire accidents are unknown. | H | H | L | V.Imp |
| Fire accident data are confidential and may not be viewed or available for research and scientific studies. | H | H | L | V.Imp |
| There is not enough staff to carry out the investigation of fire accidents. | M | M | L | V.Imp |
| The cause unknown in the fire accident investigations is acceptable to close the investigation. | H | M | L | V.Imp |
| The prevailing culture is that the task of the competent authority is to fight fires only. | M | M | L | V.Imp |
| **Contractor Attitude** | **S** | **O** | **D** | **Risk** |
| Use the cheapest products to make the most profit. | H | M | M | Important |
| The general view of fire requirements as a governmental requirement, rather than as a means to save lives. | H | M | M | Important |
| The lack of adequate control over the implementation of the requirements by the consulting firms. | H | H | M | Important |
| **Rescue speed** | **S** | **O** | **D** | **Risk** |
| The type of vehicles and equipment used by the competent authority. | H | M | H | Not Imp |
| Inadequate training of firefighters. | H | M | M | Important |
| Traffic congestion to reach residential areas. | H | H | L | V.Imp |
| Geographical distribution of fire stations in the Emirate. | H | M | L | V.Imp |
| Distribution of firefighters to fire stations. | M | M | L | V.Imp |
| Incident-reporting mechanism. | H | M | L | V.Imp |
| Fire trucks are not given priority on the road. | H | L | L | Not Imp |
| Procedures followed during the accident. | H | M | L | V.Imp |
| **Resource allocation** | **S** | **O** | **D** | **Risk** |
| Unequal distribution of workers in the centers of the competent fire authority. | M | M | L | V.Imp |

| Fire Training | S | O | D | Risk |
|---|---|---|---|---|
| The equipment used are of poor reliability. | H | M | L | V.Imp |
| Lack of continuous supply of spare parts for devices and equipment. | H | M | M | Important |
| Workers in fire fighting installation and maintenance contractors are not competent. | H | M | L | V.Imp |
| Contracts regulating the relationship between the management of the residential building and the maintenance companies of fire extinguishing systems have defects. | H | M | L | V.Imp |
| Competent authority oversight is ineffective. | H | H | M | Important |
| Contractual procedures with residents restrict entry to residential apartments to remove faults. | H | M | L | V.Imp |
| Manufacturing and design defects of the fire detection system. | H | M | L | V.Imp |
| Absence of a maintenance record for the alarm and fire fighting system. | H | H | L | V.Imp |
| Preventive maintenance history labels for fire equipment can be tampered with. | H | M | M | Important |
| **Fire equipment** | **S** | **O** | **D** | **Risk** |
| Fire extinguishing equipment is not in line with the development of fire hazards. | H | M | M | Important |
| Fire alarm systems in buildings use the Conventional type. | M | M | M | Important |
| Fire extinguishers rely on training residents to be able to use them. | H | H | M | Important |
| Not compatible with the technologies of the fourth industrial revolution | M | M | L | V.Imp |
| **Building Design** | **S** | **O** | **D** | **Risk** |
| Failure to give sufficient priority to fire fighting at the design stage. | H | M | L | V.Imp |
| Focusing on the areas of the apartments without taking into account the times and sufficient escape exits. | H | H | L | V.Imp |
| Not allocating storage rooms in the apartments, forcing residents to use escape corridors as storages. | H | H | L | V.Imp |
| Not focusing on the use of fire-insulating materials in the design stages of residential building. | H | H | L | V.Imp |
| Escape routes do not correspond to the population of the building. | H | M | L | V.Imp |
| The pumps of the fire fighting system do not correspond to the height of the residential building. | H | M | M | Important |

**Table 6.** *Cont.*

| Factor: Fire Regulations | S | O | D | Risk | Fire Training | S | O | D | Risk |
|---|---|---|---|---|---|---|---|---|---|
| There is no equality between workers in rescue centers and workers in preventive maintenance. | M | M | M | Important | Focus on reducing prices in order to reduce the safety of the population. | H | M | L | V.Imp |
| The distribution of workers between fire fighting and fire prevention centers is not based on studies, research and scientific methodologies. | M | M | M | Important | Evacuation of residents from the upper floors of high-rise residential towers is not effective. | H | M | L | V.Imp |
| Lack of workers in the centers of the competent authority. | M | M | M | Important | Failure to take into account the design, evacuating the elderly and then other people. | H | H | L | V.Imp |
| Economic measures at the level of the Government of Sharjah. | M | M | M | Important | Comparisons with successful and similar experiences in the field of designing residential building. | H | M | L | V.Imp |
| Establishing a competent local authority that affected the distribution of workers in fire fighting tasks. | M | L | M | Not Imp | Not including a control room specialized in monitoring alarm systems and surveillance cameras in residential building. | H | M | M | Important |
| Lack of information and comparisons about previous accidents | H | M | M | Important | Smoke detectors are not distributed over the entire area of the apartments. | H | M | M | Important |
| The prediction of accidents is inaccurate. | M | M | M | Important | Malfunctions in the sprinklers used in the fire extinguishing system. | H | M | L | V.Imp |
| **Government structure** | **S** | **O** | **D** | **Risk** | The gas sensor alarm is not connected to the fire alarm system. | H | H | L | V.Imp |
| The structure of fire fighting at the level of the government of Sharjah, with overlapping roles, responsibilities, and authorities. | M | M | M | Important | Failure to link the status of the fire pumps to the main alarm panel. | H | H | M | Important |
| The structure of the Sharjah government does not support the flexibility of coordination between government agencies regarding the plan of fire fighting measures. | H | M | L | V.Imp | The absence of an alarm system or fire fighting in the old residential building. | H | M | M | Important |
| The position of the Sharjah Civil Defense Authority as a local authority and its compliance with federal and local requirements may hinder efficiency and impact. | M | M | M | Important | The absence of a pressure test mechanism in the fire fighting system in the entire residential building. | H | M | M | Important |
| Fire fighting training which is supervised by another body in the government structure. | M | H | M | Important | **Combustible materials** | **S** | **O** | **D** | **Risk** |
| The early warning system, which is not directly supervised by the Sharjah Civil Defense. | H | H | L | V.Imp | The use of flammable materials in the exterior cladding of residential building. | H | H | L | V.Imp |
| The management of the early warning system by a semi-governmental company, which hinders its accountability. | H | H | L | V.Imp | The use of combustible materials in different areas of residential building when carrying out construction. | H | H | L | V.Imp |
| Absence of a national strategy to combat fire in the Emirate of Sharjah. | H | M | M | Important | The use of flammable materials by residents of residential building. | H | H | L | V.Imp |
| **Urban planning** | **S** | **O** | **D** | **Risk** | Flammable materials are not precisely defined and precautions are not taken to reduce their risks. | H | M | L | V.Imp |
| Overcrowded residential areas do not allow fire fighting vehicles to reach the accident at the required speed. | H | H | L | V.Imp | **Urbanization** | **S** | **O** | **D** | **Risk** |
| Planning parking lots around residential building hinders the work of emergency and fire fighting teams. | H | H | L | V.Imp | Residents from outside the country are not prepared to deal with the dangers of fire in residential building. | H | H | L | V.Imp |
| Concentration of high-rise buildings in specific areas. | H | H | M | Important | Immigrants from non-urban areas are causing fires because they are not aware of its dangers. | H | H | L | V.Imp |
| The closeness of the towers to each other, which threatens the possibility of the transmission of fire from one tower to another. | H | H | M | Important | Lack of knowledge of the correct procedures for dealing at the time of fire for the expatriate population from non-urban areas. | H | H | L | V.Imp |

**Table 6.** *Cont.*

| Factor: Fire Regulations | S | O | D | Risk | Fire Training | S | O | D | Risk |
|---|---|---|---|---|---|---|---|---|---|
| The narrow distance between residential buildings and the main road, which increases the risks to the residents in the event of evacuation and hinders emergency operations. | H | H | M | Important | Failure to conduct studies of fire risks resulting from the residence of expatriates from non-urban areas in high-rise buildings. | H | H | L | V.Imp |
| The lack of planning for fire stations among the public building in the city. | H | M | M | Important | Not specifying the maximum height in residential areas. | H | M | M | Important |
| Narrow roads, which impede the arrival of ambulances and fire fighting on time. | H | M | M | Important | Accommodation of state and non-urban migrant workers in multi-floored housing building. | H | M | L | V.Imp |
| **Human behavior** | **S** | **O** | **D** | **Risk** | | | | | |
| Some religious beliefs. | H | L | L | Not Imp | | | | | |
| Smoking addiction. | H | M | M | Important | | | | | |
| Improper use of electrical appliances and equipment. | H | M | L | V.Imp | | | | | |
| Cooking and grilling. | H | M | L | V.Imp | | | | | |
| Deliberately closing smoke detectors | H | M | M | Important | | | | | |
| Handling of cooking gas. | H | M | L | V.Imp | | | | | |
| Dealing with HVAC equipment. | H | M | M | Important | | | | | |
| Children's behavior | M | M | L | V.Imp | | | | | |
| The use of incense. | H | H | L | V.Imp | | | | | |

**Table 7.** Critical factors effecting the fire management in the HRBs in the Emirate of Sharjah.

| No. | Factors | Weight |
|---|---|---|
| 1. | Building design | 7% |
| 2. | Fire regulations | 8% |
| 3. | Rescue speed | 8% |
| 4. | Fire knowledge | 11% |
| 5. | Human behavior | 6% |
| 6. | Firefighting maintenance | 6% |
| 7. | Fire culture of society | 9% |
| 8. | Fire training | 6% |
| 9. | Combustible materials | 11% |
| 10. | Fire enforcement regulations | 11% |
| 11. | Accident investigation | 11% |
| 12. | Urbanization | 9% |

The failure mode, effect, and criticality analysis provide a clear and deep evaluation of the etch factor to measure the effect and possible failure mode. It is calculated based on severity, occurrence, and detection. Ten experts in the fire management system in the Emirate of Sharjah who are different stakeholders were involved in the analysis. The data collected and the result will be of added value to the Sharjah Civil Defense Authority and other stakeholders, who will use the analysis as guidance to predict failures and take the necessary preventive action to avoid fire accidents and increase the level of prevention in residential high-rise buildings in the Emirate of Sharjah.

To increase the level of possible protection, the factors classified as important were considered as critical ones, but for the purpose of this paper, only the factors classified as very important were considered to be critical factors affecting the fire management system in the high-rise building in the Emirate of Sharjah.

More analyses were carried out for the critical factors identified by failure mode, effect, and criticality analysis to determine the weight of each sub-factor through the Analytical Hierarchy Process (AHP) and use the outcome as an index to measure the effect of fire factors on the fire management system in high-rise buildings in the Emirate of Sharjah.

According to FMECA tools, the critical factors were identified, as shown in Table 7, and the top critical factors are: combustible materials, fire knowledge, fire enforcement regulations, accident investigation, urbanization, and the fire culture of the society.

The Analytic Hierarchy Process (AHP), since its invention, has been a tool at the hands of decision makers and researchers, and it is one of the most widely used multiple criteria decision-making tools [29] for the general factors affecting the Emirate of Sharjah, according to the outcomes of subject matter experts. They were analyzed using the AHP tool to identify the weight and the priority to measure the HRB's fire effect.

The AHP produces an index for the critical factors affecting high-rise buildings in the Emirate of Sharjah, as shown in Table 8.

Through the Analytical Hierarchy Process (AHP), a further analysis was carried out for the critical factors to identify the sub-critical factors and the weight of each of them to use them to build an index to measure the effect, as shown in Table 8.

The top sub-factors effecting high-rise building in the Emirate of Sharjah are: the building is fully covered with cladding, the effect of stopping activities in HRBs that are non-compliant with the fire regulations, the residents practice activities related to fire knowledge, fire regulations efficiency, the training of new employees by their employers, and the residents has fire-related knowledge, as shown in Table 9.

**Table 8.** Sub-factors affecting the fire management system in the HRBs.

| Factors | Weight | Sub-Factor | Sub-weight | Total Weight |
|---|---|---|---|---|
| Building design | 0.069 | Building in design phase | 0.209 | 0.014 |
| | | Building in construction phase | 0.248 | 0.017 |
| | | Building in use phase | 0.226 | 0.016 |
| | | Building in change phase | 0.184 | 0.013 |
| | | Building in demolition phase | 0.134 | 0.009 |
| Fire regulations | 0.077 | Legislation breakdown | 0.193 | 0.015 |
| | | Fire regulations scope | 0.268 | 0.021 |
| | | Fire regulations efficiency | 0.539 | 0.042 |
| Rescue speed | 0.079 | Compliance to the fire regulations | 0.316 | 0.025 |
| | | Distance from the fire station | 0.208 | 0.016 |
| | | Building height | 0.214 | 0.017 |
| | | Knowledge, ability, training, and experience | 0.262 | 0.021 |
| Fire knowledge | 0.106 | Beliefs related to fire knowledge | 0.362 | 0.038 |
| | | Practices related to fire knowledge | 0.411 | 0.043 |
| | | Philosophies related to fire knowledge | 0.228 | 0.024 |
| Human behavior | 0.059 | Human behavior: proactive | 0.572 | 0.034 |
| | | Human behavior: reactive | 0.257 | 0.015 |
| | | Human behavior: neutral | 0.171 | 0.010 |
| Firefighting maintenance | 0.058 | Firefighting maintenance: training | 0.106 | 0.006 |
| | | Firefighting maintenance: resources | 0.138 | 0.008 |
| | | Firefighting maintenance: integration | 0.135 | 0.008 |
| | | Firefighting system: reactive maintenance | 0.174 | 0.010 |
| | | Firefighting system: proactive maintenance | 0.257 | 0.015 |
| | | Firefighting system: predictive maintenance | 0.190 | 0.011 |
| Fire culture of society | 0.088 | Values related to fire culture | 0.200 | 0.017 |
| | | Conditions related to fire culture | 0.179 | 0.016 |
| | | Procedures related to fire culture | 0.282 | 0.025 |
| | | Behaviors related to fire culture | 0.339 | 0.030 |
| Fire training | 0.060 | Fire training: theory | 0.188 | 0.011 |
| | | Fire training: practical | 0.459 | 0.028 |
| | | Fire training: methodology | 0.353 | 0.021 |
| Combustible materials | 0.106 | The building is fully covered with cladding | 0.457 | 0.048 |
| | | The building is partial covered with cladding | 0.293 | 0.031 |
| | | The building is without cladding | 0.250 | 0.026 |
| Fire enforcement regulations | 0.106 | Fire enforcement regulations depend on fines for non-compliant facilities | 0.270 | 0.029 |
| | | Non-compliant facilities may be prohibited by fire enforcement regulations | 0.412 | 0.043 |
| | | Building owners who violate fire regulations are brought to trial | 0.318 | 0.034 |
| Accident investigation | 0.106 | Report major/minor/near-miss fire accident | 0.188 | 0.020 |
| | | Investigation major/minor/near-miss fire accident | 0.231 | 0.024 |
| | | Analysis major/minor/near-miss fire accident | 0.189 | 0.020 |
| | | Corrective actions to the cause of fire accident | 0.218 | 0.023 |
| | | Preventative actions to the cause of fire accident | 0.175 | 0.018 |
| Urbanization | 0.088 | Awareness of newcomer by the real estate companies | 0.329 | 0.029 |
| | | Training of new employees by their employers | 0.460 | 0.040 |
| | | Awareness of newcomer with visa procedures | 0.210 | 0.018 |

**Table 9.** Sub-critical factors effecting the fire management in the HRBs in the Emirate of Sharjah.

| No. | Sub-Factors | Weight |
|---|---|---|
| 1. | The building is fully covered with cladding | 5% |
| 2. | The effect of stopping activities in the HRBs that are non-compliant with fire regulations | 4% |
| 3. | Residents practices related to fire knowledge | 4% |
| 4. | Fire regulations efficiency | 4% |
| 5. | Training of new employees by their employers | 4% |
| 6. | Residents believes related to fire knowledge | 4% |
| 7. | The effect of proactive resident behavior during the fire accident | 3% |
| 8. | The effect of brought to trial HRBs owners violate fire regulations | 3% |
| 9. | The building is partially covered with cladding | 3% |
| 10. | Resident behaviors related to fire culture | 3% |
| 11. | Fire awareness of newcomer by the real estate companies | 3% |
| 12. | The effect of fines for non-compliant HRBs | 3% |
| 13. | Practical fire training | 3% |
| 14. | The effect of the building being without cladding | 3% |
| 15. | HRBs in full compliance with the fire regulations | 3% |
| 16. | Procedures implemented in the HRB by the residents related to fire culture | 2% |
| 17. | Investigation major/minor/near-miss fire accident | 2% |
| 18. | Philosophies of residents related to fire knowledge | 2% |
| 19. | Corrective actions to the cause of fire accident | 2% |
| 20. | Fire training: methodology | 2% |
| 21. | Firefighters' knowledge, ability, training and experience | 2% |
| 22. | Scope of fire regulations | 2% |
| 23. | Analysis of major/minor/near-miss fire accident | 2% |
| 24. | Report major/minor/near-miss fire accident | 2% |
| 25. | Preventative actions to the cause of fire accident | 2% |
| 26. | Awareness of newcomer with visa procedures | 2% |
| 27. | Values related to fire culture | 2% |
| 28. | Fire arrangements for the building during construction phase | 2% |
| 29. | High-rise building height | 2% |
| 30. | Building distance from the fire station | 2% |
| 31. | Conditions of the HRB effected the fire culture | 2% |
| 32. | The fire arrangements for building in use phase | 2% |
| 33. | The effect of reactive residents behavior during fire accident | 2% |
| 34. | Firefightin system proactive maintenance | 1% |
| 35. | Legislation to be breakdown (laws, regulations, standards, and guidelines) | 1% |
| 36. | Fire to be considered from building design phase | 1% |
| 37. | Fire arrangements for the building in case of change of purpose | 1% |
| 38. | Fire training: theory | 1% |
| 39. | Firefighting system predictive maintenance | 1% |
| 40. | Firefighting system reactive maintenance | 1% |
| 41. | Residents behavior to be nutral during fire accident | 1% |
| 42. | Fire arrangements for the building in demolition phase | 1% |
| 43. | Firefighting maintenance: resources | 1% |
| 44. | Firefighting maintenance: integration | 1% |
| 45. | Firefighting maintenance: training | 1% |

## 4. Discussion

### 4.1. Fire Fighting Legislation

It is necessary to develop fire fighting legislation that is in line with the development of strategies to identify the causes of fires. The causes of fires vary largely with the development of equipment used in HRB residential buildings. The Sharjah Civil Defense Authority was established as a local authority, and it needs to develop legislation that is compatible with the nature of fires in the Emirate of Sharjah, especially in relation to fires occurring in HRBs. Stakeholders must be updated on the applicable fire legislation to ensure their opinions are well informed in a changing world. The stakeholders' perspective is that the most important form of legislation to be that which effectively provides the maximum degree of protection from fire to real-estate developers, financing agencies, insurance companies, fire companies, and residents.

### 4.2. Compulsory Fire Legislation

The role of the Sharjah Civil Defense Authority is vital to ensure that all HRBs comply with fire legislation. Periodic supervision and inspection visits are carried out, requiring all HRBs to issue annual certificates of completion and penalizing HRB that have not obtained the annual certificate of compliance or that do not comply with fire fighting legislation. Converting compliance monitoring to digital monitoring contributes to the effectiveness of compliance monitoring by taking advantage of the early warning system, Aman, which links 7000 residential and commercial buildings with a unified control system, according to the statistics published by the Sharjah Prevention and Safety Authority for the year 2022. Using the techniques of the Fourth Industrial Revolution can contribute to increasing the percentage of compliance with fire legislation.

### 4.3. Management of HRBs

The safety management of HRBs is an important step to ensure the protection of residents from fire risks. Legislation requires the appointment of a fire risk officer in each facility. A fire risk management system helps the owners and the Sharjah Civil Defense Authority to reduce and control fires when they arise. The poor management of fire risks endangers the safety of the residents. Moreover, it contributes to an increase in the number of fire accidents, thus providing guidelines that help real-estate developers and owners to manage fire safety in high-rise residential buildings, which can contribute to improving the efficiency of the applied procedures.

### 4.4. Fire Research and Development

The Emirate of Sharjah is home to one of the largest universities in the UAE. The cooperation between the Sharjah Civil Defense Authority and the existing universities in the Emirate of Sharjah in the field of research and development contributes to our knowledge of the nature of fires, as well as the development of scientific approaches to fire fighting, benefiting from the latest international studies and research in the field.

### 4.5. Accident Investigation

The investigation of fire accidents provides important information to understand the nature of fires and their causes. The current investigation process in the Emirate of Sharjah is carried out at the request of the Public Prosecution for the purpose of determining the compensation procedures that will be paid by insurance companies. The Public Prosecution undertakes investigations into fire accidents, and it assigns specialized technical processes to competent authorities in relation to fire protection. To ensure access to the root cause of a fire is obtained, and to ensure that another fire does not arise again as a result of the same cause, the investigation of accidents needs to be included in the organizational structure of the Sharjah Authority for Civil Defense, and relevant employees should conduct accident investigations.

### 4.6. Contractor Attitude

Simpson describes perception as a way of seeing or understanding, attitude as a way of thinking or behaving, and behavior as a way of acting or functioning [30], and the process of installing fire fighting systems in facilities requires consulting offices to be effective in order to ensure the quality of the materials used and the accuracy and effectiveness of designs. The contractors' lack of awareness of the risks of a system failure during operations may affect the quality of the implementation of fire and alarm systems in the facilities in the Emirate of Sharjah.

### 4.7. Speed of Response and Rescue

The residents in residential facilities are in need of a response from the Sharjah Civil Defense Authority when their fire system fails to control a fire, the efforts of the staff in the residential facility fail, and the efforts of the residents also fail. In this case, the situation becomes completely out of control. A slow response from the Sharjah Civil Defense increases the human and asset losses, thus, it is essential that the speed of the response to fire disasters is considered to be one of the main performance indicators for the effectiveness of the work of the Sharjah Civil Defense Authority.

### 4.8. Optimization of Fire Resources

Some European countries, such as the United Kingdom and those in Scandinavia, have worked to implement stronger preventive measures that contribute to reducing fire accidents. The Emirate of Sharjah should similarly focus on preventive measures to reduce the occurrence of fires, such as employing more firefighters to control fires and ensuring the optimal distribution of resources.

### 4.9. Human Behavior

The fire risk in informal settlements is a function of complex interactions between the built environment, the natural environment, and people [31]. One of the most important factors in the fire fighting process is the behavior of the residents in residential facilities. Negative behaviors cause fires. Some residents have possibly fire-causing habits, such as the use of incense, which may contribute to the occurrence of fires. Negative behaviors need to be addressed via the continuous spread of awareness.

### 4.10. Fire Training

Training, in general, aims to convey knowledge and situational skills relevant to a specific context to a trainee [32], and training workers in the residential facilities on the dangers of fire contributes to removing the causes of fire, improves the procedures for dealing with fire in the event of an outbreak, and improves the emergency response operations.

### 4.11. Knowledge of Fire Hazards

Fire-related knowledge, beliefs, and practices that have been developed and applied on specific landscapes for specific purposes by long-time inhabitants [33]. It is necessary to increase the knowledge of the population about fires and their causes. Increasing the population's knowledge of the dangers of fires contributes to a reduction of the possibility of a fire being caused.

### 4.12. Culture of Society

"Safety culture" is defined as a set of values, conditions, procedures, and behaviors recognized both individually and collectively in the organization that is under consideration, regarding the organization of a management system to prevent and protect against errors, incidents, breakdowns, cyber-attacks, system integration, and accidents, and to promote safety-oriented behaviors between cooperating organizations in normal and emergency situations [34], and the culture surrounding fire in the community needs continuous improvement in order to remove the negative attitude toward the dangers of fire. The

process of educating the community is a continuous process that starts with educating school students, workers in the facilities, and the population in order to establish a positive culture that prevents the occurrence of fires.

### 4.13. Fire Fighting Technology

The technology used in fire fighting determines the effectiveness of carrying out the task of fire fighting. Using the latest and most advanced technology will contribute to building a strong fire fighting culture and reducing the rate of fires in the Emirate of Sharjah.

### 4.14. Absence or Poor Preventive Maintenance of Fire Fighting Systems

While these newer maintenance strategies require increased commitments to training, resources, and integration, there are three basic types of maintenance programs, including reactive, preventive, and predictive maintenance [35]. Maintenance is necessary to maintain the serviceability of fire fighting systems. The absence or poor quality of preventive maintenance makes the first firewall weak, and the fire fighting system may be unable to deal with fires. Continuously operating fire pumps in an automatic mode and ensuring the serviceability of the bare minimum of the systems, e.g., the water in the water tank, contribute to fighting fires that might break out in a facility. The serviceability of alarms is also important, as they ensure that residents are alerted in the event of a fire, especially in high-rise buildings.

### 4.15. Fire Fighting Equipment

The presence of complete and advanced equipment assists in fighting and controlling fires. The selection of equipment is an important element in the fire fighting process, as is the identification and testing of advanced specifications and ensuring their suitability to the environment and the nature of the Emirate of Sharjah.

### 4.16. Residential Building Design

An appropriate fire safety design should ensure occupant safety first when a building fire occurs [36]; the fire fighting process starts from the design stage of the facility using fire-resistant materials in all materials used in the construction process, providing adequate and appropriate escape exits, designing an extinguishing system that covers all parts of the facility, and reducing the possibility of a fire and reducing its effects in the event of its occurrence.

### 4.17. Flammable Materials

External cladding has been identified as a more critical component in buildings than it has been before due to many catastrophic fire incidents that have occurred in recent decades [36], but the use of external cladding in high-rise building needs more testing processes to improve the materials that are used for it, as the currently used materials are flammable when they are exposed to high temperatures. Flammable materials may be used by residents or during the construction of a residential facility. As such, the monitoring of flammable materials and prevention of their use could contribute to reducing the rate of fires in the Emirate of Sharjah.

## 5. Conclusions

According to the literature review, 19 factors affecting fire prevention management systems were identified, and they were then further classified into four categories: management factors, human factors, technical factors, and other factors. After consulting experts in the field of fire prevention management systems in the Emirate of Sharjah, 17 factors were identified, and an extra 3 factors related to the Emirate of Sharjah were added: government structure, urban planning, and urbanization. By using the FMECA tools, 12 factors were identified as critical success factors affecting the fire management systems in the Emirate of Sharjah.

The critical factors identified were analyzed by using Analytical Hierarchy Process (AHP) tools to identify the weight and priority of the sub-factors; the outcome is a list of 45 fire sub-factors affecting high-rise buildings in the Emirate of Sharjah that were identified. This is considered to be an index for Sharjah government authorities to increase the level of fire protection in high-rise buildings through the correction of the factors and sub-factors identified in this paper.

**Author Contributions:** Conceptualization, A.M.; methodology, A.M.; validation, S.B.A.A.; investigation, M.O.; data curation, M.O.; writing—original draft preparation, M.O.; writing—review and editing, S.B.A.A.; visualization, M.O.; supervision, A.M. All authors have read and agreed to the published version of the manuscript.

**Funding:** This research received no external funding.

**Informed Consent Statement:** Study did not involve humans.

**Data Availability Statement:** The data presented in this study are available on reasonable request from the corresponding author.

**Conflicts of Interest:** The authors declare no conflict of interest.

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
