# Peer review of "Critical Factors Affecting Fire Safety in High-Rise Buildings in the Emirate of Sharjah, UAE"

_fire, doi:10.3390/fire6020068_

Round 1
Reviewer 1 Report
This paper presents critical factors affecting fire safety in High-Rise Buildings in 2 the Emirate of Sharjah, UAE 3, but there are significant issues on the data process and analysis. An extensive revision should be done before could be considered for publication.
1. The Abstract should focus on the contributions and conclusions of this paper, and it is suggested to reorganize the abstract.
2. The abstract is on the short side, it is suggested to expand the word count.
3. The innovation of this paper is not clear in Introduction section, and it is suggested to reorganize or strengthen the innovation description.
4. The methods and quantity of literature review should be described in detail.
5. This paper selects 16 experts to analyze the influencing factors, whether the number of experts is reasonable
6. The form of Figures needs to be improved in this paper.
Author Response
Dear: reviewer
The major review conducted in the paper to meet the desired requirements and ensure that all of your notes are corrected is listed in comments in the word file for conformation.
Best Regards

Reviewer 2 Report
1. As this is a typical case study, the author should describe specifically the features or fire saftey (potential risk) in the Emirate of Sharjah, UAE.
2. Introduction needs to be much improved with relating to high rise building fires.
3. The authors should emphasise the feasibility of using an expert perspective to evaluate fire factors in this paper
4. Language needs to be more precise and less colloquial
Author Response

(The authors gave the same response as above.)

Reviewer 3 Report
The authors address a very important topic of fire safety in high rise buildings. Even though the study is conducted only for structures in Emirate of Sharjah, some of findings can be applicable to any high-rise structure in general. It is a well written article but following points need to be addressed;
1. Abbreviations need to be defined either in a separate nomenclature section or where first used.
2. Introduction feels incomplete and disconnected. Needs to be reworked.
3. The Literature review section feels more like a motivation section. It is supposed to provide references to similar studies done in the past, highlighting some shortcomings in them.
4. What does figure 1 represent? It is a percentage of what?
5. Table 1 is very confusing and needs further explanation.
6. Better explanation is required on how Table 4 was constructed. For example it is not clear why “Fire enforcement regulations” which has 100% events flagged as ‘Very Important’ is placed on second spot whereas ‘Fire regulations’ which has only 73% events under ‘Very Important’ placed on number 1 spot.
7. English language requires some attention throughout the article. For example, in the Conclusion section "After apply FMECA tools" should be "After application of the FMECA tools".
Author Response

(The authors gave the same response as above.)

Round 2
Reviewer 1 Report
This paper has been modified according to the reviewer's comments and it is suggested that this paper could be accepted in present form.
Reviewer 2 Report
The paper in this version can be accepted for publication and all the comments are appropriately addressed.